# Non-collinear magnetic structures in the magnetoelectric Swedenborgite $CaBaFe_4O_7$ derived by powder and single-crystal neutron diffraction

Navid Qureshi[1,2]⋆, Bachir Ouladdiaf[1], Anatoliy Senyshyn[3], Vincent Caignaert[4] and Martin Björn Valldor[5,2]

**1** Institut Laue-Langevin, Grenoble, France
**2** *II*. Physikalisches Institut, Universität zu Köln, Germany
**3** Forschungs-Neutronenquelle Heinz Maier-Leibnitz (FRM-II), Technische Universität München, Garching, Germany
**4** CRISMAT, UMR 6508, CNRS-ENSICAEN, Caen, France
**5** Centre for Materials Science and Nanotechnology (SMN), Department of Chemistry, University of Oslo, Norway

⋆ qureshi@ill.fr

## Abstract

We have investigated the magnetic structures of the Swedenborgite compound $CaBaFe_4O_7$ using magnetic susceptibility and neutron diffraction experiments on powder and single-crystal samples. Below $T_{N1} = 274$ K the system orders in a ferrimagnetic structure with spins along the *c* axis and an additional antiferromagnetic component within the kagome plane which obviously cannot satisfy all exchange interactions. Competing single-ion anisotropy and exchange interactions lead to a transition into a multi-q conical structure at $T_{N2} = 202$ K. The derivation of the complex ordering scheme below $T_{N2}$ is an important step towards the understanding of the magnetoelectric effect under magnetic fields in this polar ferrimagnet.

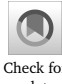

# 1 Introduction

Geometrical frustration [1,2] occurs in lattices of vertex-sharing triangles, e.g. kagome layers [3,4] or pyrochlore nets [5,6], in which the antiferromagnetic exchange interactions of nearest neighbours cannot be satisifed. Crystal structures with a high degree of frustration, e.g. a network of equilateral triangles, may not reveal a long-range ordered magnetic ground state even down to very low temperatures. However, small distortions from the high-symmetry crystal structures allow the spin system to order in interesting and exotic ways.

The magnetic Swedenborgites, with its first member reported as $YBaCo_4O_7$ in 2002 [7], are structural homologues to the hexagonal mineral $SbNaBe_4O_7$ [8,9], and are extensively studied due to their interesting crystal structures and diverse magnetic properties. The first observations from the magnetic lattice of hexagonal $YBaCo_4O_7$, i.e. diffuse neutron scattering on powder, suggested only short range spin order [7]. By diluting its magnetic lattice with a non-magnetic ion in $YBa(Co_{4-x}Zn_x)O_7$ ($x = $ 0-3) the properties gradually change into a spin-glass [7,10]. In the orthorhombically distorted Swedenborgite $YbBaCo_4O_7$, a long-range order was indicated with sharp Bragg reflections in neutron diffraction experiments, however, only the propagation vectors could be identified and not the full spin structure [11]. Simultaneously, it became obvious that the oxygen stoichiometry was important for the resulting symmetry of the atomic lattice in the Swedenborgites [12]. Additionally, the single-ion anisotropy (magnetocrystalline) effects apparently affect the symmetry and magnetism. In $CaBaCo_4O_7$, having a $Co^{2+}/Co^{3+}$ ratio of 1, the atomic lattice is orthorhombic and a ferrimagnetic-like ground state was reported [13,14]. In the subject of this study - $CaBaFe_4O_7$, with a similar charge composition - the Swedenborgite lattice is also orthorhombically distorted and a long-range spin order appears already close to room temperature [15,16]. Magnetocurrent measurements revealed the magnetoelectric effect under the application of an external magnetic field for which a non-collinear and non-coplanar spin order was claimed to be responsible [17], therefore making the link to the highly interesting material class of multiferroics [18,19]. However, no neutron diffraction study devoted to the details of the involved magnetic structure exists in the literature, which is the main motivation for the investigation presented below.

The actual, magnetic ground states in the Swedenborgite systems strongly depend on the type of structural distortion away from the hexagonal symmetry, which releases the geometric frustration to some degree and allows for a magnetic state with several, similarly strong, but competing spin interactions. Also, with the uneven distribution of electrons among the $d$ orbitals in tetrahedral crystal fields in the Swedenborgite structure, contributions from Jahn-Teller-like single-ion anisotropy would be valid for $d^6$ ions like $Fe^{2+}$ and $Co^{3+}$, of which the former is present here. However, with the present data it will not be possible to exclusively relate the magnetic properties with the local electric phenomena, but those investigations could be very important for the future understanding of spin-ordering phenomena in Swedenborgites.

According to the Mermin-Wagner [20] theorem, a 1D or 2D isotropic spin-S Heisenberg model cannot reveal long-range order. Although magnetocrystalline anisotropy is expected for the $Fe^{2+}$ ($d^6$) ion [the $Fe^{3+}$ ($d^5$) spin is symmetrical in tetrahedral crystal fields], a significant magnetic coupling between the kagome layers is probably an important factor for the mag-

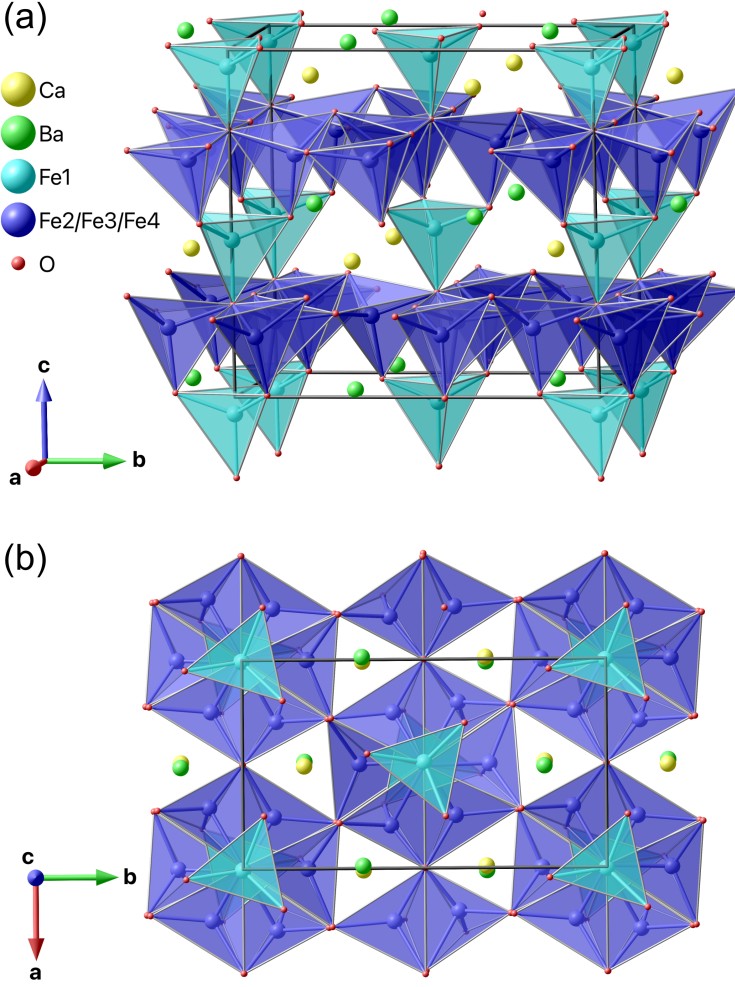

Figure 1: (a) Visualization of the crystal structure of $CaBaFe_4O_7$ consisting of triangular Fe sites (light blue) and hexagonal Fe sites (dark blue), where the latter form the kagome planes within the $a$-$b$ plane. (b) View along the $c$ axis emphasizing the close-to hexagonal symmetry and the two different types of triangles within the kagome planes as described in the text.

netic properties in Swedenborgites. Therefore, the triangular layer of magnetic ions between the kagome layers, see Figure 1(a) for the case of $CaBaFe_4O_7$, plays a decisive role for the appearance of long-range order in these systems as it can mediate the spin-spin interaction leading to a 3D spin system.

By viewing the crystal structure along the $c$ axis [Figure 1(b)] it can be seen that each kagome layer reveals two different types of Fe triangles: the first type of triangle ($T_1$) is situated around the vertical connection between two Fe spins of the triangular plane, while the second type of triangles ($T_2$) surrounds either a Ca or Ba cation.

In the present study, neutron diffraction experiments on powder and single-crystal samples reveal the magnetic structures of the $CaBaFe_4O_7$ compound which offer an interesting insight into the exchange couplings between the planes and especially within the two different types of triangles of the kagome planes.

## 2  Experimental

The synthesis of powders and growth of single crystalline $CaBaFe_4O_7$ is described in detail elsewhere [16]. In short, single crystals (> 1 cm) were grown in an optical floating-zone furnace. Pieces of the single crystal were ground into powder to assure that all data, presented here, correspond to the same sample.

The magnetic susceptibility measurements were done with a vibrating sample magnetometer (VSM, 40 Hz, 2mm) in a physical property measurement system (PPMS, Quantum Design) by cooling under an applied magnetic field of $\mu_0 H = 1$ T. All susceptibility data shown here were taken from [16]. Powder neutron diffraction data [21] were obtained at SPODI (FRM II, Munich, Germany) [22], using a constant wavelength of 2.537 Å. About 20 grams of sample powder were placed in a sample holder of vanadium and the cryostat walls were all of aluminum. Helium was used as cooling agent in a top-loading closed-cycle refrigerator from Vericold. Diffraction patterns were recorded at 15 K and 300 K as well as in 15 K steps between 105 K and 270 K. The neutron single-crystal diffraction experiment [21] was carried out at the D10 diffractometer (ILL, Grenoble) in the four-circle geometry. A single-crystal specimen of 3x3.5x4 mm$^3$ (along the $a$, $b$ and $c$ axes) was used. The nuclear structure was investigated using two different wavelengths, one being $\lambda_1 = 2.36$ Å employed from the (002) reflection of a HOPG monochromator and the other $\lambda_2 = 1.26$ Å from the (200) reflection of a Cu monochromator. All integrated intensities were corrected for absorption applying the transmission factor integral $\exp[-\mu(\tau_{in} + \tau_{out})]$ by using MAG2POL [23] ($\tau_{in}$ and $\tau_{out}$ represent the path lengths of the beam inside the crystal before and after the diffraction process, $\mu$ is the linear absorption coefficient, which is 0.0056 mm$^{-1}$ for $CaBaFe_4O_7$ at $\lambda_1$ and 0.0096 mm$^{-1}$ at $\lambda_2$, respectively).

The powder diffraction data were analyzed using the FULLPROF [24] package, while all single-crystal diffraction data were treated with MAG2POL [23].

## 3  Results

### 3.1  Single-crystal measurements

#### 3.1.1  Nuclear structure

We have investigated the nuclear structure at RT by collecting 722 and 119 symmetry-inequivalent reflections (1541 and 996 unique reflections) at $\lambda_1$ and $\lambda_2$, respectively. Apart from two scale factors, one for each data set, the refined parameters were the atomic positions, the isotropic temperature factors (constrained to be equal for same elements on different sites) and the diagonal elements of the extinction correction tensor within an empirical SHELX-like model [25]. The refinement returned acceptable agreement factors of $R_{F,1} = 10.9$ and $R_{F,2} = 5.9$ for the two data sets with $\lambda_1$ and $\lambda_2$, respectively.

Since the orthorhombic Swedenborgite crystal structure is very closely related to the undistorted hexagonal structure of $SbNaBe_4O_7$ and the $CaBaFe_4O_7$ compound reveals a trigonal symmetry at higher temperatures [26], we have repeated the structural analysis by including 3 orthorhombic twins being rotated by 120° as shown in Figure 2 and by refining their populations.

The inclusion of twins reveals a significant improvement of the refinement quality, which is expressed by $R_{F,1} = 4.7$ and $R_{F,2} = 2.9$, and the presence of a perfectly twinned sample with homogeneously distributed twins. The refined parameters are shown in Table 1.

Table 1: Refined nuclear structure parameters within the $Pbn2_1$ space group at RT ($R_{F,1} = 4.7$, $R_{F,2} = 2.9$, $\chi^2 = 4.3$). The only Wyckoff site in this space group is the general $4a$ site. Note that not all atomic positions can be refined at the same time due to the absence of a special position, i.e. the origin needs to be fixed. The extinction parameters $x_{ii}$ are the diagonal entries of a tensor used to calculate the extinction factor. Note that the isotropic temperature factor $B$ has been constrained to be the same for elements on different sites.

| Atoms | $x$ | $y$ | $z$ | $B$ (²) |
|---|---|---|---|---|
| Ca | 0.011(3) | 0.6686(6) | 0.8915(8) | 0.69(8) |
| Ba | 0.001(2) | 0.6696(5) | 0.5203(9) | 1.39(5) |
| Fe1 | 0.001(2) | 0.000(2) | 0.9516(8) | 0.82(1) |
| Fe2 | 0.003(2) | 0.1782(2) | 0.6997(8) | 0.82 |
| Fe3 | 0.2935(5) | 0.0934(3) | 0.1941(9) | 0.82 |
| Fe4 | 0.2471(5) | 0.9139(4) | 0.7007(8) | 0.82 |
| O1 | 0.001(2) | 0.003(2) | 0.2665(8) | 1.05(2) |
| O2 | 0.004(2) | 0.5007(3) | 0.2562(9) | 1.05 |
| O3 | 0.7835(8) | 0.2633(5) | 0.8053(9) | 1.05 |
| O4 | 0.7180(7) | 0.7531(6) | 0.2244(9) | 1.05 |
| O5 | 0.054(1) | 0.1565(4) | 0.514(1) | 1.05 |
| O6 | 0.1958(9) | 0.1102(5) | 0.019(1) | 1.05 |
| O7 | 0.2508(9) | 0.9402(4) | 0.516(1) | 1.05 |

Lattice parameters
$a = 6.3135$ Å $\quad b = 11.0173$ Å $\quad c = 10.3497$ Å
Extinction parameters
$x_{11} = 0.005(2) \quad x_{22} = -0.0005(3) \quad x_{33} = 0.0013(1)$
Twin populations
twin 1: 0.337 $\quad$ twin 2: 0.328(7) $\quad$ twin 3: 0.335(8)

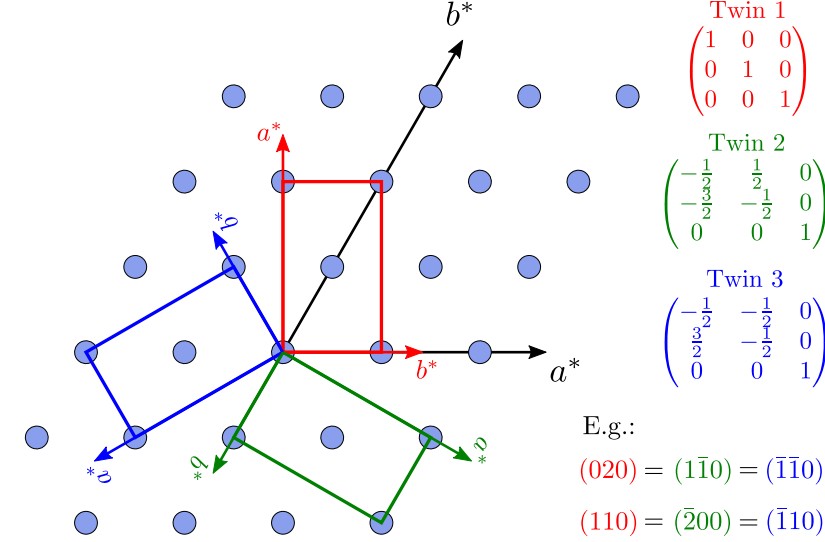

Figure 2: Sketch of the reciprocal space showing 3 twins rotated by 120° as a consequence from a high-temperature structural transition from a trigonal to an orthorhombic structure. The actually observed scattering vectors **Q** are obtained by multiplying the twin matrices by the nominal **Q** vector of twin 1.

### 3.1.2 Magnetic phase transitions

Figure 3(a) shows the susceptibility curves as a function of temperature for a magnetic field of $H = 1$ T applied either parallel or perpendicular to the $c$ axis of the Swedenborgite structure. As this strong magnetic anisotropy already appears far above the first magnetic ordering temperature, i.e. in the paramagnetic range, it might be argued that single-ion anisotropy is present in the system. $Fe^{2+}$ ($d^6$ ion) in a tetrahedral crystal field obviously allows for a local preferred orientation of its magnetic spin. However, without further data, it is only possible to speculate on how significant this contribution is to the spin ordering phenomenon. At $T_{N1} = 274$ K a local maximum is visible in the $H \perp c$ curve, while the $H \parallel c$ curve reveals a large increase of $\chi$ upon cooling indicative of a ferro- or ferrimagnetic structure with magnetic moments along the $c$ axis with an additional antiferromagnetic component perpendicular to $c$. The anomaly at $T_{N2} = 202$ K visible only in the $H \parallel c$ curve suggests a spin reorientation of the in-plane component.

The integrated intensities of selected Bragg reflections from the single-crystal neutron diffraction experiment are depicted in Figure 3(b) on the same temperature scale. Clear anomalies coincide with the transition temperatures observed in the magnetic susceptibility. On cooling through $T_{N1}$ a strong increase of intensity is seen in the (020) and (110) reflections, while only a moderate increase is present in the (002) reflection. Since only the perpendicular component of the ordered magnetic moment with respect to the scattering vector $\mathbf{Q}$ contributes to magnetic scattering the intensity evolution suggests a predominant alignment of the spins parallel to the $c$ axis with a smaller in-plane component, in perfect agreement with the interpretation of the susceptibility curves. At $T_{N2} = 204$ K the (002) reflection - being sensitive only to the in-plane component - reveals a drop in intensity at the same temperature at which additional satellite reflections - modulated by a propagation vector $\mathbf{q} = (1/3\ 0\ 0)$ - appear. This suggests that the in-plane component breaks translation symmetry upon cooling through $T_{N2}$. The absence of any clear anomaly in the integrated intensities of the (020) and (110) reflections indicate that the $c$ component of the magnetic moments is not affected at this transition.

### 3.1.3 Magnetic structures

For the determination of the magnetic structure between $T_{N1}$ and $T_{N2}$ 114 symmetry-inequivalent reflections (696 unique reflections) were recorded at $T = 220$ K. Due to the relatively large temperature difference between the magnetic and nuclear data collection the analysis was done by refining the nuclear and magnetic structure parameters simultaneously. The twin model shown in Fig. 2 was employed with the populations fixed to the values obtained from the RT structure analysis. Symmetry analysis was employed to derive magnetic structure models being compatible with the underlying crystal structure and the propagation vector $\mathbf{q} = 0$. This task was done using the MAG2POL program and the 4 different irreducible representations are shown in Table 2. From the basis vectors one can deduce that only $\Gamma_2$ yields a ferromagnetic component along the $c$ axis within a single Fe site ($w$ coefficients positive for all 4 atomic positions), while revealing an antiferromagnetic coupling of the components $u$ and $v$ within the $a$-$b$ plane (2 positive and 2 negative $u/v$ coefficients). Nevertheless, all models were tested on the observed data, but only $\Gamma_2$ returned a good agreement. The parameters $u$, $v$ and $w$ were constrained to be of the same size for the 3 Fe sites within the kagome plane. This is a reasonable assumption based on the XMCD results in [16] stating that the Fe magnetic moment at the trigonal sites (Fe1) is larger than those in the kagome planes (Fe2-4), meaning that the latter are closer to $Fe^{2+}$. In a first refinement step the $a$ component proved to be insignificant for all 4 sites and was set to 0 in the following. Furthermore, the refinement procedure was very sensitive to the $b$ component, so its absolute value was constrained between the Fe sites in the triangular and kagome planes. This constraint stabilized the refine-

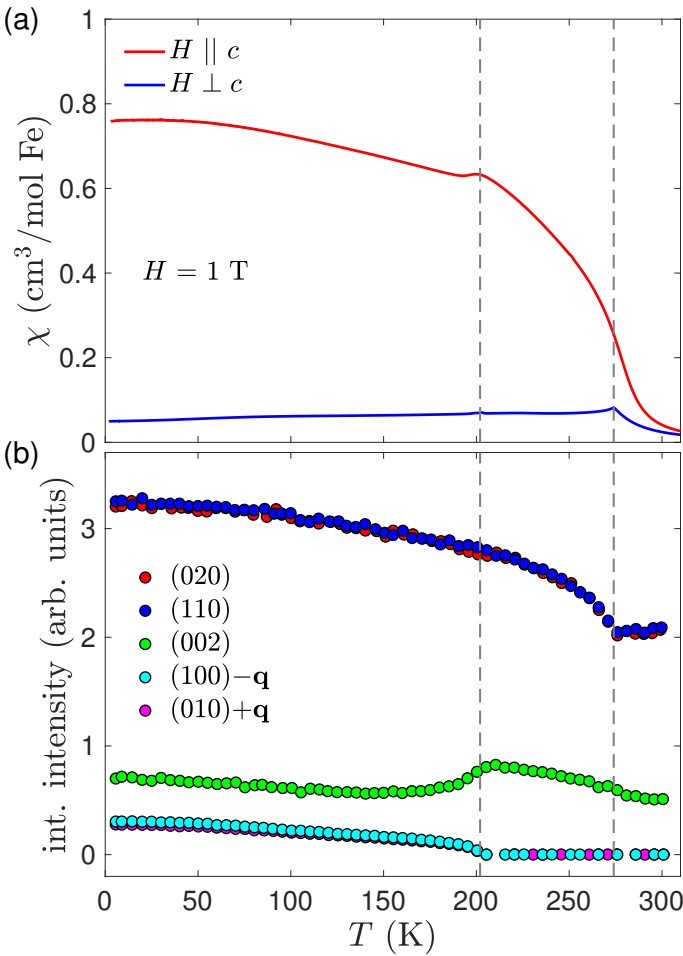

Figure 3: (a) Magnetic susceptibility measurements in a field of $\mu_0 H = 1$ T on single crystalline $CaBaFe_4O_7$ plotted against temperature. The curves for $H \perp c$ and $H \parallel c$ (taken from [16]) reveal a local maximum indicating the magnetic phase transitions at $T_{N1} = 274$ K and $T_{N2} = 202$ K, respectively (marked as vertical dashed lines). (b) Integrated intensities for selected integer $(hkl)$ and satellite Bragg peaks from the D10 experiment. The anomalies in the temperature dependence correspond exactly to the magnetic phase transition temperature in (a). The evolution of the respective Bragg peak intensities allow a very good guess of the involved magnetic structures as described in the text.

ment and the agreement factor $R_F = 4.8$. The resulting magnetic structure can be described as a ferrimagnetic configuration with $\mu \parallel c$ between the Fe spins in the triangular planes and those in the kagome planes, where the larger moment of the Fe1 ion is in agreement with the aforementioned distribution of $Fe^{2+}$ (Fe2-4, kagome) and $Fe^{3+}$ (Fe1, trigonal). Furthermore, an antiferromagnetic canting of the spins is present along the $b$ axis, which creates the classic situation of not being able to satisfy all antiferromagnetic exchange interactions on a triangle, i.e. 2 parallel and 1 antiparallel spin. The resulting magnetic structure is shown in Figure 4 and the refined values are shown in Table 3. It has to be noted that a solution with a slightly worse agreement factor exists, in which the $b$ component is uniform within a single kagome plane. However, such a model with satisfied ferromagnetic in-plane exchange interactions would not lead to the second magnetic phase transition observed at $T_{N2}$.

A slightly reduced data set of Bragg peaks with integer indices has been recorded within the low-temperature phase at $T = 2$ K with 119 symmetry-inequivalent reflections (202 unique).

Table 2: Basis vectors $\psi_n$ of the irreducible representation $\Gamma_n$ for each of the Fe sites of $CaBaFe_4O_7$ for space group $Pbn2_1$ and propagation vector $\mathbf{q} = (0\ 0\ 0)$.

| Atom | Position | $\psi_1$ | $\psi_2$ | $\psi_3$ | $\psi_4$ |
|---|---|---|---|---|---|
| 1 | $\begin{pmatrix} x \\ y \\ z \end{pmatrix}$ | $\begin{pmatrix} u \\ v \\ w \end{pmatrix}$ | $\begin{pmatrix} u \\ v \\ w \end{pmatrix}$ | $\begin{pmatrix} u \\ v \\ w \end{pmatrix}$ | $\begin{pmatrix} u \\ v \\ w \end{pmatrix}$ |
| 2 | $\begin{pmatrix} \bar{x} \\ \bar{y} \\ z+1/2 \end{pmatrix}$ | $\begin{pmatrix} \bar{u} \\ \bar{v} \\ w \end{pmatrix}$ | $\begin{pmatrix} \bar{u} \\ \bar{v} \\ w \end{pmatrix}$ | $\begin{pmatrix} u \\ v \\ \bar{w} \end{pmatrix}$ | $\begin{pmatrix} u \\ v \\ \bar{w} \end{pmatrix}$ |
| 3 | $\begin{pmatrix} \bar{x}+1/2 \\ y+1/2 \\ z \end{pmatrix}$ | $\begin{pmatrix} u \\ \bar{v} \\ \bar{w} \end{pmatrix}$ | $\begin{pmatrix} \bar{u} \\ v \\ w \end{pmatrix}$ | $\begin{pmatrix} u \\ \bar{v} \\ \bar{w} \end{pmatrix}$ | $\begin{pmatrix} \bar{u} \\ v \\ w \end{pmatrix}$ |
| 4 | $\begin{pmatrix} x+1/2 \\ \bar{y}+1/2 \\ z+1/2 \end{pmatrix}$ | $\begin{pmatrix} \bar{u} \\ v \\ \bar{w} \end{pmatrix}$ | $\begin{pmatrix} u \\ \bar{v} \\ w \end{pmatrix}$ | $\begin{pmatrix} u \\ \bar{v} \\ w \end{pmatrix}$ | $\begin{pmatrix} \bar{u} \\ v \\ \bar{w} \end{pmatrix}$ |

Table 3: Refined magnetic parameters of the magnetic structure at 220 K and of the commensurate spin component at 2 K. The components $\mu_b$ and $\mu_c$ correspond to the refined parameters $v$ and $w$ shown in Table 2. The numbering of Fe atoms is analogous to Table 1.

| | $T = 220$ K | | $T = 2$ K | |
| Atom | $\mu_b$ ($\mu_B$) | $\mu_c$ ($\mu_B$) | $\mu_b$ ($\mu_B$) | $\mu_c$ ($\mu_B$) |
|---|---|---|---|---|
| Fe1 | 1.0(2) | 3.1(1) | 0.2(9) | 3.68(8) |
| Fe2 | 1.0(2) | 2.2(1) | 0.2(9) | 2.84(7) |
| Fe3 | 1.0(2) | 2.2(1) | 0.2(9) | 2.84(7) |
| Fe4 | 1.0(2) | 2.2(1) | 0.2(9) | 2.84(7) |

The same refinement strategy was applied as for the $T = 220$ K data set, i.e. refining the nuclear structure parameters as well as the magnetic structure components $v$ and $w$ within irreducible representation $\Gamma_2$. We observe an increase of the $c$ component due to the reduced temperature as well as an insignificant $b$ component (see Table 3), which confirms the assumption of a modulated in-plane component. As the refinements of both nuclear and magnetic structures turn out satisfactory, there seems to be no need of introducing a $Fe^{2+}/Fe^{3+}$ charge ordering with accompanying Fe-O bond-length modulations.

As a last step of the single-crystal experiment 1314 magnetic satellites were collected that agree with the propagation vector $\mathbf{q} = (1/3\ 0\ 0)$ at $T = 2$ K. Symmetry-compatible magnetic structure models were again calculated using MAG2POL which are shown in Table 4. Unfortunately, neither a single irreducible representation nor any mixed representation yielded a satisfying result. This is due to the fact that nuclear scattering from additional twin domains overlap with parts of the magnetic scattering. This is manifest by multiple diffraction spots on the 2-dimensional detector images and multiple peaks in the $\omega$ scans which are impossible to resolve and to separate into individual contributions. Note that such parasitic scattering was not observed in the rocking scans of integer reflections. It is therefore not possible to confidently extract the magnetic intensities and to analyze the modulated part of the low-

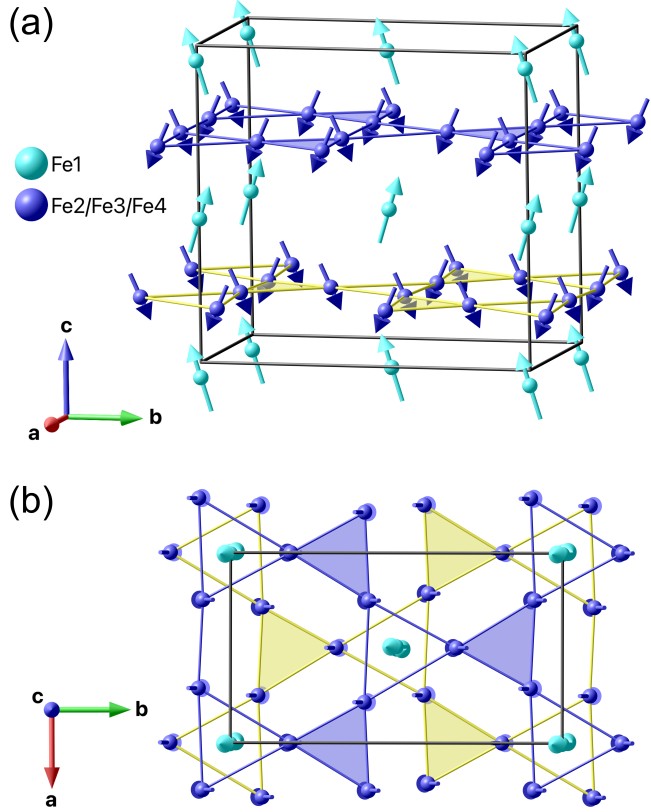

Figure 4: (a) Perspective view of the magnetic structure in $CaBaFe_4O_7$ at 220 K. Only the magnetic ions on the triangular (light blue) and hexagonal sites (dark blue) are shown. Bonds between Fe ions in the kagome planes are drawn as a guide to the eye. The kagome plane at $z \sim 0.2$ is emphasized in yellow in order to be distinguished from the one at $z \sim 0.7$. $T_1$ triangles are open, while $T_2$ triangles are filled. (b) View along the $c$ axis emphasizing the $b$ component of the magnetic moments. All Fe triangles in the kagome plane reveal 2 spins pointing along the positive (negative) $b$ axis, while 1 spin is pointing along the negative (positive) $b$ axis.

temperature magnetic phase from our single-crystal data. A polarized neutron approach using spherical neutron polarimetry - as employed for the related $CaBa(Co_3Fe)O_7$ compound [27], failed due to the strong ferrimagnetic component throughout the whole magnetically ordered temperature range despite the effort of prior cooling in a magnetic field (in order to reduce neutron depolarization between magnetic domains) and focusing only on incident and final neutron polarization states parallel to the ferrimagnetic component (longitudinal polarization analysis).

## 3.2 Powder neutron diffraction

Due to the difficulties in deriving the low-temperature in-plane component encountered in the single-crystal experiment we now turn to our powder neutron diffraction data in order to address this remaining issue. The sequence of magnetic phase transitions coincides with the results above which is shown in the following.

All recorded diffraction patterns between 15 K and 300 K were used to construct the thermodiffractogram depicted in Figure 5. The transition into the canted ferrimagnetic structure at $T_{N1}$ is marked by the intensity increase of commensurate reflections e.g. at scattering angles 26.8°,

Table 4: Basis vectors $\psi_n$ of the irreducible representation $\Gamma_n$ for each of the Fe sites of $CaBaFe_4O_7$ for space group $Pbn2_1$ and propagation vector $\mathbf{q} = (1/3\ 0\ 0)$. Note that each of the Fe sites splits into two orbits. The phase factor $a = \exp(2\pi i \mathbf{q} \mathbf{r})$ results from the $n$ glide plane perpendicular to the $b$ axis with translation vector $\mathbf{r} = (1/2\ 0\ 1/2)$.

| Atom | Position | $\psi_1$ | $\psi_2$ |
|---|---|---|---|
| 1 | $\begin{pmatrix} x \\ y \\ z \end{pmatrix}$ | $\begin{pmatrix} u \\ v \\ w \end{pmatrix}$ | $a \cdot \begin{pmatrix} \bar{u} \\ v \\ \bar{w} \end{pmatrix}$ |
| 2 | $\begin{pmatrix} x + 1/2 \\ \bar{y} + 1/2 \\ z + 1/2 \end{pmatrix}$ | $\begin{pmatrix} u \\ v \\ w \end{pmatrix}$ | $a \cdot \begin{pmatrix} u \\ \bar{v} \\ w \end{pmatrix}$ |

$30.4°$ and $39.4°$. The onset of the modulated phase at $T_{N2}$ is accompanied by the appearance of magnetic satellites from which the strongest are located at $2\theta = 15.4°$ and $21.0°$. Note that the positions of the satellites do not change with temperature. A few selected Bragg reflections at positions with integer and non-integer Miller indices were integrated using a Gaussian profile on a sloping background in all diffraction patterns which were used to construct the color map in Figure 5. The resulting temperature dependence of integrated intensities is shown in Figure 6. The first transition, at $T_{N1}$, can only be interpreted as a jump of the (020), (110) (at $2\theta = 26.7°$ and $26.8°$, respectively) and (111) ($2\theta = 30.4°$) intensities between 300 K and 270 K due to the lack of recorded data within this temperature range. The integrated intensites of the satellites (010)+$\mathbf{q}$ ($2\theta = 15.4°$) and (101)+$\mathbf{q}$ ($2\theta = 21.0°$) show a significant increase below 210 K. Both transition temperatures match very well with the more detailed picture shown in Figure 3(b) derived from the single-crystal sample.

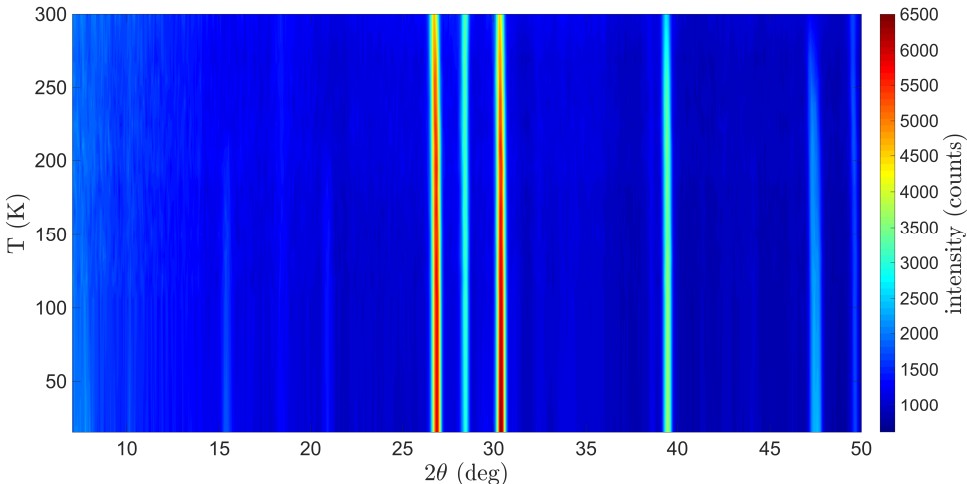

Figure 5: Thermodiffractogram showing the magnetic phase transitions at $T_{N1} = 274$ K and $T_{N2} = 204$ K (note that the temperature values were derived from the single-crystal experiments). The onset of the commensurate ferrimagnetic structure is manifest by an increase of intensity on e.g. the reflections at $2\theta$ values of $26.8°$, $30.4°$ and $39.4°$. The transition into the low-temperature magnetic phase is accompanied by the appearance of new satellite peaks, e.g. at $2\theta = 15.4°$ and $21.0°$.

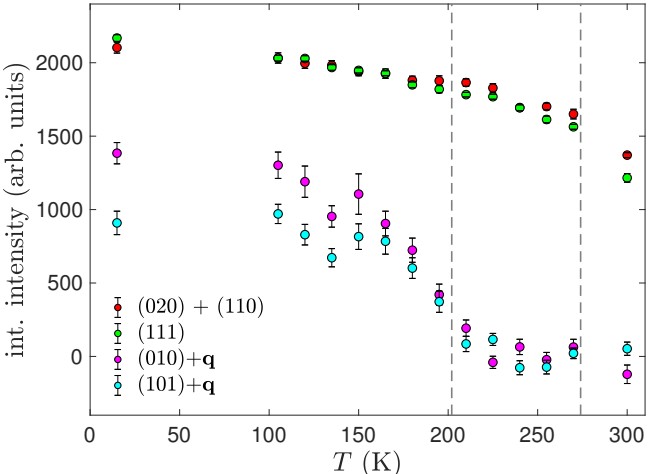

Figure 6: Integrated intensities of selected Bragg reflections at positions with integer and non-integer Miller indices, the latter being moduated by the propagation vector $\mathbf{q} = (1/3\ 0\ 0)$. The (020) and (110) reflections cannot be separated due to their very similar scattering angle ($2\theta = 26.7°$ and $26.8°$, respectively), but both reveal a significant magnetic contribution which results in a comparable temperature dependence as the single peak (111) at $2\theta = 30.4°$. The magnetic satellites show an increase in intensity upon cooling below approximately 200 K. The transition temperatures derived from the single-crystal experiments (cf. Fig. 3) are shown as vertical dashed lines.

As a first step the diffraction pattern at RT was analyzed in order to refine the nuclear structure parameters, an overall isotropic temperature factor and the scale factor. The observed pattern can nicely be described using the known structure ($R_F = 8.2$) which is shown in Figure 7(a). The resulting structural model was used as a starting point for the analysis of the 15 K pattern. The scale factor was left unchanged and only the lattice parameters and the overall isotropic temperature factor were refined in order to guarantee the correct position and scaling of the magnetic satellites. The propagation vector was refined to $\mathbf{q} = [0.3354(5)\ 0\ 0]$. Figure 7(b) zooms on the low-$Q$ part of the diffraction pattern containing the clearly visible magnetic satellites. Apart from the two strongest magnetic Bragg peaks already visible in the thermodiffractogram the relatively weak fundamental reflection (000)+$\mathbf{q}$ can be seen at $2\theta = 7.7°$ as well as a series of peaks between $32°$ and $45°$. The strong nuclear reflections as well as parasitic peaks observable at all temperatures (e.g. at $10.3°$ and $18.4°$ in $2\theta$) were excluded from the refinement.

The irreducible representations listed in Table 4 were used, however, the complexity of the nuclear and magnetic structure in combination with the limited number of observed magnetic reflections requires reasonable constraints and starting parameters to assure refinement stability. Since the Fe sites split into two orbits due to the reduced propagation vector symmetry and each site features an $a$ and $b$ component as well as a phase factor, the maximum number of magnetic structure parameters is 23 (note that the phase of one Fe site needs to be fixed). Therefore, as already applied in the analysis of the high-temperature magnetic phase the size of the $a$ and $b$ component was constrained to be the same for Fe spins on the same type of site, i.e. within the triangular or kagome planes. As a starting point of the refinement process different classical spin configurations on a kagome lattice were introduced on the Fe triangles in the kagome plane - including 120° spin arrangements on the $T_1$ and/or $T_2$ triangles - by fixing the respective phase factors, which were then refined within either $\Gamma_1$, $\Gamma_2$, $\Gamma_1 + \Gamma_2$ symmetry or without symmetry constraints.

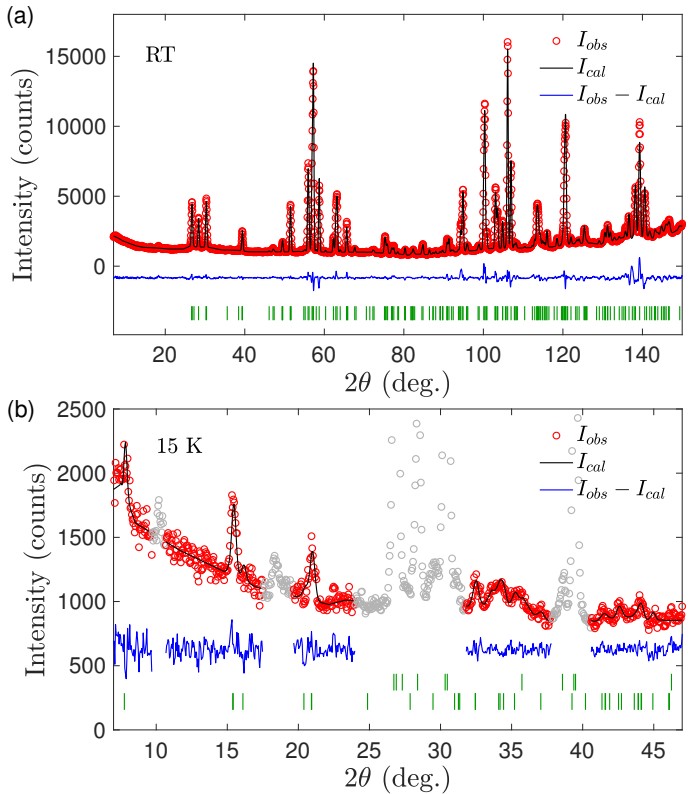

Figure 7: Observed [(red) dots] and calculated [(black) solid line] diffraction patterns at (a) RT and (b) 15 K with the difference curve shown (in blue) at the bottom. In (a) the (green) markers indicate the position of nuclear Bragg peaks within the $Pbn2_1$ space group. In (b) the first row of (green) markers denotes the position of nuclear Bragg peaks, while the second row indicates the positions of magnetic Bragg peaks with the propagation vector $\mathbf{q}=(1/3\ 0\ 0)$. Gray data points show the regions which were excluded from the fit for containing either nuclear peaks or parasitic peaks also present above the magnetic ordering temperatures.

A very convincing solution was found by constraining only the $T_1$ triangles to reveal a 120° spin arrangement within $\Gamma_1$ symmetry. The phase factors between two triangles separated along the $z$ axis as well as between the triangular Fe spins were refined together with the spin envelope in the $a$-$b$ plane for triangular and kagome sites. After the first refinement steps the $a$ and $b$ components of both Fe types revealed similar values for which the spin envelope was constrained to be circular reducing the total number of refinable parameters to 5. We obtain an agreement factor of $R_F = 12.7$ and the good agreement between the calculated and observed patterns can be seen in Figure 7(b), the refined parameters are listed in Table 5. The circular spin envelope with an amplitude of 1.6 $\mu_B$ at 15 K matches very well with the collinear $b$ component of 1.0 $\mu_B$ which was determined at an elevated temperature of $T = 220$ K.

Apart from the same spin envelope for all sites it is obvious that the refined phase factor between the Fe2' and Fe2 spin is close to $2\pi/3$ and the one between Fe1' and Fe1 is almost zero. Therefore, in principle, the magnetic structure could be described with only 2 free parameters, which are an overall moment amplitude and the phase factor between the kagome and triangular planes. Such a minimal model still yields $R_F = 13.7$ compared to $R_F = 12.7$ with 5 parameters. The commensurate component along the $c$ axis together with the cycloidal component within the $a$-$b$ plane results in a conical magnetic structure which is depicted in Figure 8 and will be discussed in the following section.

Table 5: Refined magnetic parameters of the modulated in-plane magnetic structure component at 15 K ($R_\mathrm{F} = 12.7$). The numbering of Fe atoms is analogous to Table 1 and the positions of the primed Fe atoms are related to the unprimed ones by the $b$ glide plane lost in the transition.

| Atom | $\mu_a$ ($\mu_\mathrm{B}$) | $\mu_b$ ($\mu_\mathrm{B}$) | $\varphi/(2\pi)$ |
|------|------|------|------|
| Fe1  | 1.6(3) | 1.6(3) | 0 |
| Fe1' | 1.6(3) | 1.6(3) | 0.04(3) |
| Fe2  | 1.6(1) | 1.6(1) | 0.09(3) |
| Fe2' | 1.6(1) | 1.6(1) | 0.77(3) |
| Fe3  | 1.6(1) | 1.6(1) | $\varphi(Fe2') + 1/3$ |
| Fe3' | 1.6(1) | 1.6(1) | $\varphi(Fe2) + 1/3$ |
| Fe4  | 1.6(1) | 1.6(1) | $\varphi(Fe2) - 1/3$ |
| Fe4' | 1.6(1) | 1.6(1) | $\varphi(Fe2') - 1/3$ |

## 4 Conclusion

We have presented a combination of magnetic susceptibility and neutron diffraction experiments on powder and single-crystal samples which address the magnetic phases in the $CaBaFe_4O_7$ compound and reveal yet another type of magnetic ordering adding to the rich diversity of examples within the Swedenborgite family. All employed techniques reveal two magnetic phase transitions, the first at $T_{\mathrm{N}1} = 274$ K into a ferrimagnetic structure with antiferromagnetic canting perpendicular to the easy direction, and the second at $T_{\mathrm{N}2} = 202$ K where the in-plane component changes from a collinear to a cycloidal arrangement which results in a conical magnetic structure at low temperatures. This sequence of magnetic phase transitions is an excellent example of the temperature-dependent competition between single-ion anisotropy and exchange interactions. In the high-temperature phase the collinear $b$ component creates the textbook situation of two parallel and one antiparallel spins on a triangle, the prototypic example of geometric frustration. Between 274 K and 202 K the spin Hamiltonian seems to be dominated - at least for the in-plane component - by the single-ion anisotropy which reduces the system's energy by canting the spins along the $b$ axis. However, when the temperature is lowered the frustrated antiferromagnetic exchange interaction become more important for which a spin reorientation takes place towards a partial 120° arrangement. The in-plane component of this complex structure can be appreciated in Figure 8(b) by viewing it along the $c$ axis. One can see that the same 120° spin configuration is present on two $T_1$ triangles, above as well as below a triangular Fe spin. Apart from the antiferromagnetic coupling within each of those triangles such a structure suggests a ferromagnetic exchange interaction between two triangular plaquettes along the $c$ axis. This seems to be the decisive characteristic of the magnetic structure, because a spin configuration which yields a 120° alignment on all triangles - which does not explain the experimental data - requires an opposite triangular chirality between two $T_1$ triangles separated by $z \sim 0.5$. Consequently, the $T_2$ triangles do not show an apparent coupling scheme for which we conclude that the exchange interactions within those triangles play a minor role in the spin Hamiltonian of this Swedenborgite compound. The structural origin of the different ordering schemes between $T_1$ and $T_2$ triangles presumably lies in the vicinity of the triangular Fe spins which cap the $T_1$ triangles above and below, which therefore leads to a different balance of exchange interactions. The bare presence of a canted ferrimagnetic order is proof for a strong coupling between the planes, and a cluster consisting of ferromagnetically ordered $T_1$ triangles with apparent 120° order within the plaquettes above and below a triangular Fe spin suggests that the resulting magnetic structure is governed by the superexchange interactions within these units. In contrast, the

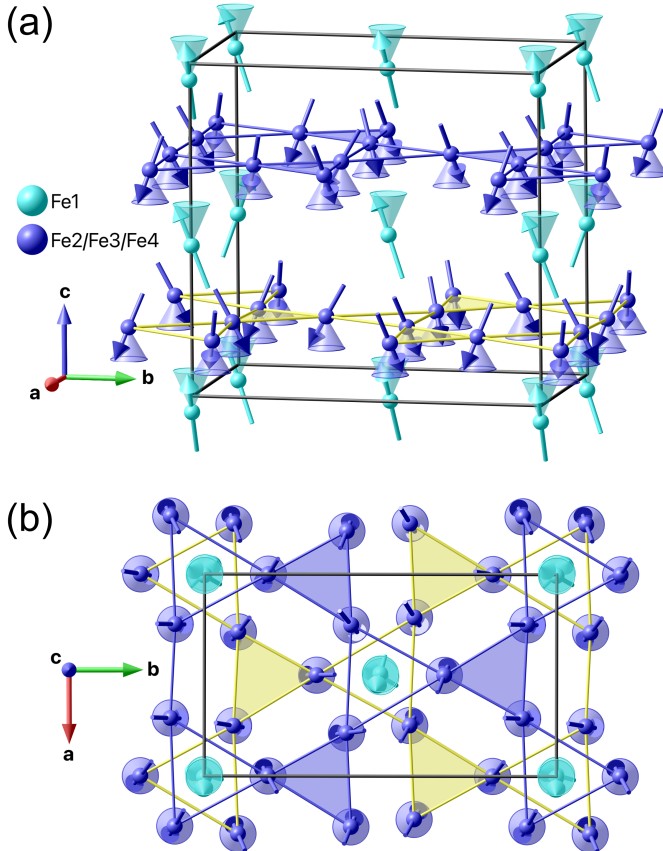

Figure 8: (a) Perspective view of the conical ferrimagnetic structure in $CaBaFe_4O_7$ at 15 K. Only the magnetic ions on the triangular (light blue) and hexagonal sites (dark blue) are shown. The conical envelope of the magnetic moments as well as bonds between Fe ions in the kagome planes are drawn as a guide to the eye. The kagome plane at $z \sim 0.2$ is emphasized in yellow in order to be distinguished from the one at $z \sim 0.7$. $T_1$ triangles are open, while $T_2$ triangles are filled. (b) View along the $c$ axis emphasizing the rotation of the magnetic moments within the $a$-$b$ plane. The spin rotation plane is emphasized by disks in the respective colors. The triangular plaquettes $T_1$ of kagome Fe spins reveal a 120° configuration. The same spin orientation and triangular chirality is found for the triangle at $\Delta z = 0.5$ indicating a ferromagnetic coupling between two plaquettes.

magnetic interactions between the clusters (note that a $T_2$ triangle constitues the intersection of 3 clusters) are not perfectly fulfilled and - in turn - are less dominant in the energy balance, which may be related to the fact that $T_2$ triangles are structurally more isolated due to the absence of another $T_2$ triangle along the $c$ direction.

On the other hand it is not quite clear why the system reveals a small in-plane component besides the strong ferrimagnetic component along the $c$ axis and how the low-temperature magnetic structure is responsible for inducing a ferroelectric polarization when applying a magnetic field. Whether the microscopic origin of this near-room-temperature multiferroic is magnetostriction, the spin-current mechanism or Fe-O orbital hybridization, as put forward by Kocsis *et al.* [17], is still an open debate. The precise Fe-O-Fe bond distances and angles between the triangular and kagome layers, as well as within the $T_1$ and $T_2$ triangles, as a function of temperature would certainly reveal valuable information about this remaining question, but this is beyond the possibilities of the data at hand. Further investigations, e.g.

using high-resolution X-ray synchrotron diffraction, are required to reveal the structural origin of the observed magnetic structures. For a more precise picture of the energy balance in the spin Hamiltonian additional inelastic neutron scattering studies would be necessary based on the structural and magnetic properties provided in this work. Nevertheless, the details of the complex magnetic order at low temperatures combined with the magnetoelectric data [17] may stimulate further *ab initio* calculations in order to provide a solid base for the understanding of the magnetoelectric effect in this system.

# Acknowledgements

**Funding information** This work was supported by the German Science Foundation (DFG) through SFB608 and SFB1143.

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
