# Peer review of "Non-collinear magnetic structures in the magnetoelectric Swedenborgite CaBaFe4O7 derived by powder and single-crystal neutron diffraction"

_SciPost Physics Core, doi:SciPost Phys. Core 5, 007 (2022)_

## Round 1 · Referee Report · Anonymous (Referee 1) · 2021-10-2

Strengths

Extensive neutron study of complex magnetic structures in a potentially interesting frustrated magnetic material.

Weaknesses

This work falls short of explaining the microscopic origin of magnetic order in CaBaFe4O7. It raises (and does not answer) multiple questions on which interactions and anisotropies are important, and how they can be rationalized with respect to the crystal structure of this material.

Report

The authors report on the magnetic structure determination for CaBaFe4O7. This swedenborgite-type compound is potentially interesting by virtue of its coexisting triangular and kagome planes and a somewhat non-trivial magnetic structure, which has not been resolved to date. That being said, the compound was disclosed back in 2011 by some of the authors [PRB 83, 180405(R)], and the data shown in the present work seem to go back to the same time, because they are actually cited as Ref. 10 of that PRB'2011 manuscript. I appreciate that collecting neutron data on single crystals and powders may be a lengthy job compared to standard lab experiments, but the time span of 10 years is still hard to justify even for a thorough neutron study. This long delay with the publication indicates to me only a marginal interest in this compound both by the authors and by the community alike. Indeed, since 2011 only two more publications have appeared (PRB 93, 014444 (2016) and PRM 2, 054403 (2018)), and surprisingly, none of them is even cited in the present manuscript!

This work is of excellent technical quality and certainly merits publication. On the other hand, it falls short of making a compelling case for SciPost Physics with regard to the acceptance criteria of the journal. The authors conclude their manuscript with a fairly scarce outlook where they claim that their results will "serve as valuable input for further dynamical and theoretical studies". However, such a statement could apply to essentially any work on the magnetic structure determination. It was not clear to me in which way the insights provided by this magnetic structure will be useful to understand peculiarities of the magnetoelectric response reported in PRB'2016, or how they fundamentally (on the level of individual magnetic interactions and anisotropies) distinguish this Fe-based system from its Co-based sister compounds. Moreover, the discussion of magnetic couplings in Section 4 seems rather superficial (see below) and does not provide sufficient information that a theoretical study could build upon.

Overall, I believe that not simply a revision but a significant extension of this work (for example, by an ab initio calculation of magnetic interactions and anisotropies) would be required to meet the acceptance criteria of SciPost Physics. On the other hand, the manuscript seems to be in a rather good shape to be published in SciPost Phys. Core or Physical Review B. I would fully support such a publication as soon as the following points are addressed:

Requested changes

  1. In my opinion, the main peculiarity of this magnetic structure lies in the coexisting FM and AFM orders for the out-of-plane and in-plane spin components, respectively. Where do the ab-components come from? Why do they even appear if the spins tend to point along 'c'? The crystal structure is a derivative of the trigonal one, so I would expect a leading anisotropy term that chooses either an in-plane or the out-of-plane spin direction. Why does it not happen? How does the trigonal-orthorhombic distortion affect the crystal structure and the local environment of the Fe atoms? Is the Jahn-Teller effect at play? I understand that the data at hand may not be sufficient to address these questions, but the authors could at least formulate them and lay down explicit directions for further research.

  2. Likewise, I could not understand where the difference between the T1 and T2 triangles comes from. Why do the T1 triangles show the 120-deg order for the ab-component of the spin, whereas the T2 triangles do not? Is it due to a difference between the triangles themselves, or due to the fact that T1's are capped with the Fe1 sites from the triangular layers? Which feature of the crystal structure is mainly responsible for the magnetic structure observed in the present experiments? Is it a deformation of the kagome planes, or the mutual stacking of the kagome planes and triangular layers?

  3. In Section 4, the authors also argue that the "weaker Dzyaloshinskii-Moriya interaction... certainly plays a role". I could not understand the reasoning behind this statement. One does not need the DM interaction to form the 120-deg order. The DM term will only choose the chirality of this order. However, it was not clear from the manuscript whether the symmetry of the DM terms on the T1 triangles is in agreement with the experimental magnetic chirality. Is it possible that the interlayer interactions (to Fe1) stabilize same chirality of the T1 triangles above and below Fe1, and the DM term is not even needed?

  4. Please, explain how you treated magnetic reflections from the different twin domains in the 220 K refinement. Was the twin model from the RT refinement used with the fixed populations of the twin domains?

  5. Please, give a proper credit to the PRB'2016 and PRM'2018 publications. The latter reports the trigonal high-T crystal structure, which you refer to as "presumably hexagonal" (Sec. 3.1.1).

  6. Please, label the T1 and T2 triangles in one of the figures.

  7. Figures 4b and 7b would be easier to understand if only the ab-spin components were shown.

  8. It would be helpful if mcif-files with the refined magnetic structures were included in the eventual publication. Readers will benefit from the direct access to the magnetic structures without the need to decipher the irreps.

  9. In the Introduction, I stumbled upon the statement "Due to the Heisenberg-like nature of the involved spins" that seems very misleading in the light of the magnetization data shown in Fig. 3a. The huge uniaxial anisotropy displayed by these data (and also by the field-dependent measurements in Fig. 2 of PRB'2016) reflects strong deviations from the Heisenberg model. I suggest that the statement in the Introduction should be revised accordingly.

  10. I was wondering why none of the data sets have been referenced in the manuscript. Is it in line with the ILL data policy?

---

## Round 1 · Referee Report · Anonymous (Referee 2) · 2021-10-9

Report

The authors present new powder and single-crystal neutron diffraction data for the Swedenborgite compound CaBaFe$_4$O$_7$. Below $T_{N_1}$ = 274 K the system is found to order in a ferrimagnetic structure with spins along the $c$ axis and an additional antiferromagnetic component within the kagome planes. A further transition into a multi-q conical structure is found at $T_{N_2}$ = 202 K.

From the experimental point of view, this seems to be solid work. However, I have a problem related to the last sentence of the abstract: "The resulting ordering schemes offer valuable insight into the coupling mechanisms which serve as valuable input for further dynamical and theoretical studies of this complex system". Indeed, CaBaFe$_4$O$_7$ seems complex if not to say complicated, and I did not see a crisp message that would attract a big audience. Even worse, the work does not seem to be put properly into context. On the formal side, not only is the reference list very short (15 references), but more than half of these are publications by the present authors. Content-wise, it may be Ok to start with references for the concrete compound, but references for the general highly-frustrated magnetism context are missing, i.e., the first three sentences of the Introduction do make some comments on highly frustrated magnets, but without any references.

More on the level of detail, the magnetic susceptibility reproduced from Ref. [13] in Fig. 3(a) is remarkably anisotropic even in the high-temperature paramagnetic regime. This highlights the importance of single-ion anisotropy. While the latter is emphasized elsewhere in the manuscript, I think it would be useful to also discuss it in the context of Fig. 3(a).

At the end of section 3.1, I was surprised that the authors did not succeed to solve the inverse problem for single crystals while it works for polycrystals. I saw that there is some discussion on this point (related to twinning), but I failed to get the message. Can the authors maybe explain further? Is the single-crystal data maybe in some respects of lower quality than the poly-crystalline one (less of reciprocal space covered or such)?

Finally, I have a problem with the color shading in Fig. 5 since I see no clear signatures of phase transitions beyond a fade-out of certain reflections. Scans such as those shown in Fig. 3(b) are really much clearer such that I think that at least Fig. 3(b) should also be mentioned in the discussion of Fig. 5.

Requested changes

This work needs to be placed properly into context: 1- Is there a crisp message that one could convey also with title and/or abstract? 2- More than half of just 15 references being work by the present authors seems off balance. 3- Cite reviews on highly frustrated magnets and/or related systems (such as kagome ones) in the Introduction.

Further details: 4-Add comments on single-ion anisotropy to the discussion of Fig. 3(a). 5- Improve discussion/clarity of Fig. 5 (e.g., by including Fig. 3(b) in the discussion). 6-Typeset chemical formulas in the titles of Refs. [3-7,9-13] correctly (see, e.g., https://tex.stackexchange.com/questions/10772/bibtex-loses-capitals-when-creating-bbl-file for typesetting hints).

---

## Round 2 · Referee Report · Anonymous (Referee 2) · 2021-12-12

Report

With their revised version, the authors have addressed most of the previous concerns. I therefore recommend it for publication in SciPost Physics Core.

Apart from some minor typographic corrections mentioned below, I stumbled across three points when rereading the manuscript that the authors might still reconsider: 1- The statement "One can immediately realize ..." on lines 174-177 at the beginning of section 3.1.3 is not obvious to me. I trust that it is correct, but the authors might wish to expand or at least reformulate. 2- I am grateful to the authors for the new Fig. 6 and the related explanations. This confirms that one can identify peak positions, but no sharp transition as a function of temperature. The authors might thus wish to reconsider the caption of Fig. 5 in order to make it clear that the precise values of $T_{\rm N1}$ and $T_{\rm N2}$ come from elsewhere. 3- Lines 332-335, first sentence of second paragraph of Conclusion: For an Ising model, one would expect an up-up-down spin configuration for each triangle while for a (classical) Heisenberg model, it should be a 120$^\circ$ configuration. So, a configuration where spins point mostly along the $c$ axis and have a small 120$^\circ$ in-plane component seems to be what one would expect for a system with a large but finite single-ion anisotropy. I understand that this is not exactly what is observed. Nevertheless, from the high-temperature behavior of $\chi$, one should be able to estimate the single-ion anisotropy $D$ as well as the nearest-neighbor Heisenberg exchange $J$ and thus get at leas a rough idea what local spin configuration one would expect for such a model. It might therefore be useful to make the data of Fig. 3(a) available.

Requested changes

1- Reconsider the statement "One can immediately realize ..." on lines 174-177 at the beginning of section 3.1.3. 2- Reconsider the caption of Fig. 5. 3a- Reconsider the first sentence of second paragraph of the Conclusion on lines 332-335. 3b- Consider making the data for $\chi$ in Fig. 3(a) available. 4- Inconsistent grammar in the second line of the caption of Table 1 ("Wyckhoff siteS ... IS"). 5- Second line of caption of Table 3: remove full stop after "component". 6- Line 193: remove full stop in "Figure. 4". 7- Second line of caption of Table 5: "analog" is fine in American English, but if the authors meant to write British English (see, e.g., "neighbours" on line 42), they should use "analogous". 8- Lines 284/285: "almost insignificant" sounds a bit strange (I think "insignificant" or "almost zero" would work). 9- Refs. [22-24] should be cited between Ref. [16] and Ref. [17]. 10- Double check casing in the references ("Mineral" [8], "K" [15], "Heisenberg" [17], ...).

---

## Round 2 · Referee Report · Anonymous (Referee 1) · 2021-12-14

Report

The authors have fully addressed my previous criticisms. I gratefully recommend publication in SciPost Physics Core, provided that two further comments are addressed:

  1. Introduction: I did not understand why only minor magnetocrystalline anisotropy is expected for Fe2+ in the tetrahedral coordination. Orbital moment of Fe2+ is not quenched, so the anisotropy is likely. Moreover, the authors themselves resort to the single-ion anisotropy in the Conclusions when they discuss the magnetic structure between TN1 and TN2. This inconsistency may confuse readers and should be avoided.

  2. Fig. 8b: I could not see the "opposite triangular chirality between two T1 triangles separated by z~0.5", or perhaps I do not quite understand what the authors mean by the "triangular chirality" (is it scalar chirality? or vector chirality?). In my view, the spins rotate counter-clockwise when going around the triangle in the clockwise direction, and this kind of chirality is the same for all the T1 triangles shown in this figure. A clarification along these lines would be helpful.

---

## Round 2 · Author Response

First of all, we would like to thank both referees for their thorough review and close inspection of our manuscript, which has helped to improve its quality.
We have addressed all of the referees' proposed development points and comments, which we will list below (the page and line numbers refer to the resubmitted manuscript).

---

## Round 2 · List of Changes

Requested changes

First referee

  1. In my opinion, the main peculiarity of this magnetic structure lies in the coexisting FM and AFM orders for the out-of-plane and in-plane spin components, respectively. Where do the ab-components come from? Why do they even appear if the spins tend to point along 'c'? The crystal structure is a derivative of the trigonal one, so I would expect a leading anisotropy term that chooses either an in-plane or the out-of-plane spin direction. Why does it not happen? How does the trigonal-orthorhombic distortion affect the crystal structure and the local environment of the Fe atoms? Is the Jahn-Teller effect at play? I understand that the data at hand may not be sufficient to address these questions, but the authors could at least formulate them and lay down explicit directions for further research.

We have rewritten the introduction (page 2, lines 47-82) and focused more on the compositions that are truly related to the Swedenborgite in question. The reviewer´s comment on the Jahn-Teller effect is probably very important to understand the reason for having spin-ordering in Swedenborgites, in general. However, as the reviewer states, we do not have the data at hand to do this, but the importance of the local electronic configuration is now mentioned in the introduction as future investigation possibility. Similarly, we have mentioned the unknown structural origins for the magnetic structure as an outlook in the Conclusion section (page 16, line 332 – 342), which has been restructured.

  1. Likewise, I could not understand where the difference between the T1 and T2 triangles comes from. Why do the T1 triangles show the 120-deg order for the ab-component of the spin, whereas the T2 triangles do not? Is it due to a difference between the triangles themselves, or due to the fact that T1's are capped with the Fe1 sites from the triangular layers? Which feature of the crystal structure is mainly responsible for the magnetic structure observed in the present experiments? Is it a deformation of the kagome planes, or the mutual stacking of the kagome planes and triangular layers?

Without further high-resolution structural, theoretical or dynamical approaches we can only speculate about what the origin of the magnetic structure is. From our point of view the stacking of kagome planes and triangular layers leading to T1 triangles which are capped by Fe1 sites and T2 triangles which are not (as correctly pointed out by the referee) is the decisive aspect. This leads to a spin configuration within and between T1 triangles which can be expressed using clear interaction schemes. The T2 triangles – being more isolated due to absence of another T2 triangle along the c direction – presumably reveal exchange interactions which are playing a minor role in the energy balance. We tried to underline this structural difference by describing T1 triangles connected by a Fe1 atom as clusters with dominating magnetic exchange interactions. T2 triangles constitute in fact only the edges of 3 different clusters and, hence, the irregular ordering schemes within those triangles can be interpreted as less dominant exchange interactions which are consequently not perfectly fulfilled. We have added a paragraph to section 4 dealing with these arguments (page 16, line 320 – 331).

  1. In Section 4, the authors also argue that the "weaker Dzyaloshinskii-Moriya interaction... certainly plays a role". I could not understand the reasoning behind this statement. One does not need the DM interaction to form the 120-deg order. The DM term will only choose the chirality of this order. However, it was not clear from the manuscript whether the symmetry of the DM terms on the T1 triangles is in agreement with the experimental magnetic chirality. Is it possible that the interlayer interactions (to Fe1) stabilize same chirality of the T1 triangles above and below Fe1, and the DM term is not even needed?

We agree with the referee that the DM interaction is not necessary to form the 120-deg order and that mentioning it in the present context is misleading. However, we are afraid that revealing the microscopic driving forces for the presented magnetic order requires either (a) an in-depth theoretical analysis including symmetric exchange couplings, spin-orbit coupling in a single triangle and eventually a DM correction within a single triangle and between triangles or (b) an inelastic neutron scattering study including the above mentioned interactions. Both are beyond the scope of the present manuscript and in order to avoid speculations we decided to remove the sentence mentioned by the referee.

  1. Please, explain how you treated magnetic reflections from the different twin domains in the 220 K refinement. Was the twin model from the RT refinement used with the fixed populations of the twin domains?

This is exactly how the 220 K data was analyzed. The same twin model was used and the populations were fixed to the values obtained from the RT structure analysis. We have added a sentence at the beginning of section 3.1.3 in order to clarify this point (page 7, lines 170-171).

  1. Please, give a proper credit to the PRB'2016 and PRM'2018 publications. The latter reports the trigonal high-T crystal structure, which you refer to as "presumably hexagonal" (Sec. 3.1.1).

This comment is indeed a down-grading of the nice works, mentioned by the reviewers. We regret our wording, therefore we have removed “presumably” and added the latter reference to that place in the text (page 4, line 130-131). Many thanks for this correction. We gave proper credit to PRB’2016 by referring to the magnetoelectric properties throughout the text.

  1. Please, label the T1 and T2 triangles in one of the figures.

Figures 4 and 8 were modified in order to distinguish between T1 and T2 triangles (T2 triangles are now filled, while T1 triangles are not), which was also added to the figure captions. In this context, we corrected two errors: 1) It is the plane at z ~ 0.2 which is emphasized in yellow, whereas the plane at z ~ 0.7 is shown in blue. 2) The refined phase factors were not correctly represented in the depicted structure due to different definitions between the program used for the refinement and the program used for the visualization. Nevertheless, all descriptions, discussions and conclusions in the text are still valid.

  1. Figures 4b and 7b would be easier to understand if only the ab-spin components were shown.

We understand the referee’s point. However, we would prefer to depict the magnetic structures in 4b and 8b as they are, which bears the advantage to be able to compare them with the perspective view in 4a and 8a, respectively (we have therefore applied the same color code and the labelling of T1/T2 triangles in the perspective view). We believe that the a-b component is sufficiently clear despite the view along c in combination with a non-zero c component.

  1. It would be helpful if mcif-files with the refined magnetic structures were included in the eventual publication. Readers will benefit from the direct access to the magnetic structures without the need to decipher the irreps.

Mcif-files have been generated and can be included in the publication for the q=0 magnetic structure at 220 K and for the q=(1/3 0 0) component at low temperature. Note that mcif-files may be useful for q=0 magnetic structures, but non-zero propagation vectors are not well handled by visualization softwares, because (a) they might not even support the propagation vector formalism and (b) the mcif-file contains real magnetic moments (and not their complex Fourier coefficients which are necessary to represent the modulation along q). In general, only the magnetic moments within the first unit cell are correctly depicted.

  1. In the Introduction, I stumbled upon the statement "Due to the Heisenberg-like nature of the involved spins" that seems very misleading in the light of the magnetization data shown in Fig. 3a. The huge uniaxial anisotropy displayed by these data (and also by the field-dependent measurements in Fig. 2 of PRB'2016) reflects strong deviations from the Heisenberg model. I suggest that the statement in the Introduction should be revised accordingly.

We agree with the reviewer that this might need further explanation – and the manuscript text is improved accordingly (page 2, lines 79-82). The involved magnetic ions, Fe2+ and Fe3+, have electron distributions that allow for close to symmetrical interactions with their surrounding. Fe2+ is d6 and Fe3+ is d5, of which only the former has degenerate electronic ground states in a tetrahedral crystal field. Hence, we agree that a part Ising like anisotropy is possible for the Fe2+ ion. However, the extra electron resides in an e-orbital and should be less involved in the magnetocrystalline anisotropy. That is why we characterize the spins as Heisenberg-like in the introduction. As the reviewer probably knows, we would need a strong Ising type spin to have a 2D long range AF-magnetic ordering (as indirectly stated by the Mermin-Wagner theorem). Now, we have rephrased that text part – being more exact in the definition – and we added one reference.

  1. I was wondering why none of the data sets have been referenced in the manuscript. Is it in line with the ILL data policy?

The data were measured before the ILL data policy was introduced. It is not possible to create a DOI label in that case.

Second referee

  1. Is there a crisp message that one could convey also with title and/or abstract?

We have modified the title and the last sentence of the abstract in light of the magnetoelectric properties of this system. As an outlook in the discussion, we have related the complex low-temperature spin order with the complementary macroscopic data of Kocsis et al., which - combined with future ab initio studies - offer a solid starting point to understand the microscopic origin of the ferroelectric polarization under applied magnetic fields.

  1. More than half of just 15 references being work by the present authors seems off balance.

We have rewritten the introduction and focused more on the compositions that are truly related to the Swedenborgite in question. During that process, several of the references have been replaced or removed. We hope this now balances the work of others with ours.

  1. Cite reviews on highly frustrated magnets and/or related systems (such as kagome ones) in the Introduction.

We have added reviews on geometric frustration, kagome lattices and pyrochlore nets in the opening sentence of the introduction providing the interested reader an easier access to the relevant literature. We have also added reviews on multiferroics at the corresponding place.

  1. Add comments on single-ion anisotropy to the discussion of Fig. 3(a).

We have added a minor comment to the text that references to Fig. 3a, and that text contains, as suggested by the reviewer, a hint that single-ion anisotropy could be a relevant factor for the spin ordering, but the statement is left open for later investigations (page 5, lines 140-145).

  1. Improve discussion/clarity of Fig. 5 (e.g., by including Fig. 3(b) in the discussion).

In order to improve the clarity of Fig. 5 we have integrated a selection of peaks with integer and non-integer Miller indices using a Gaussian profile on a sloping background at all temperatures. We have added the new Fig. 6 showing this temperature dependence of integrated intensities which we also compare to Fig. 3(b) in the text (page 11, lines 234-244) and in the figure caption.

6.Typeset chemical formulas in the titles of Refs. [3-7,9-13] correctly (see, e.g., https://tex.stackexchange.com/questions/10772/bibtex-loses-capitals-when-creating-bbl-file for typesetting hints).

The chemical formulas in the titles have been corrected.

Further minor changes

1) In relation to the second referee’s comment concerning why it was not possible to solve the magnetic structure from the single-crystal data we have added a bit of text at the end of section 3.1. In the submitted version we had already explained that the 2D detector images were “polluted” by parasitic peaks from additional twin domains (the nuclear nature is evident due to the strong contribution even at high Q values). The resulting omega scans therefore contained multiple peaks and we have added that these peaks “are impossible to resolve and to separate into individual contributions” (page 11, line 214). The reason for the “low quality” is therefore not the reciprocal space coverage, but the inability to extract the magnetic intensities. We have tried to at least estimate the nuclear and magnetic contribution to those “mixed” peaks using polarized neutrons and spherical neutron polarimetry with the Cryopad device, but the strong ferrimagnetic component throughout the whole temperature range of magnetic order rendered this task impossible. Even prior magnetic field cooling with the goal of reaching a magnetic monodomain in order to reduce neutron depolarization between magnetic domains and focusing only on initial and final neutron polarization states parallel to the ferrimagnetic component (effectively doing longitudinal polarization analysis) did not allow to extract any meaningful data. We have added a sentence concerning these attempts (page 11, lines 218-223).

2) In Table 5 the phase factor of the Fe4’ atom is phi(Fe2’)-1/3 and not +1/3.

---

## Round 3 · Author Response

We would like to thank the referees again for the close inspection of the manuscript.
We have incorporated the requested changes into the resubmitted version and we have created a DOI for the neutron data on an external server. We have furthermore added the DOIs for the references, where possible.

---

## Round 3 · List of Changes

REFEREE 1:

1- Reconsider the statement "One can immediately realize ..." on lines 174-177 at the beginning of section 3.1.3. We have reformulated this sentence which now better explains how to read the basis vectors in Table 2.

2- Reconsider the caption of Fig. 5. It is now mentioned that the values are the ones derived from the single-crystal experiments.

3a- Reconsider the first sentence of second paragraph of the Conclusion on lines 332-335. We prefer to leave the mentioned sentence as it is in order to avoid more speculations. The question of the in-plane component arises even between TN1 and TN2, i.e. not only for the complex low-temperature phase, and with the data at hand we cannot address this issue adequately.

3b- Consider making the data for χ in Fig. 3(a) available. We can make the data available upon request.

4- Inconsistent grammar in the second line of the caption of Table 1 ("Wyckhoff siteS ... IS"). This has been corrected.

5- Second line of caption of Table 3: remove full stop after "component". This has been corrected.

6- Line 193: remove full stop in "Figure. 4". This has been corrected.

7- Second line of caption of Table 5: "analog" is fine in American English, but if the authors meant to write British English (see, e.g., "neighbours" on line 42), they should use "analogous". This has been corrected.

8- Lines 284/285: "almost insignificant" sounds a bit strange (I think "insignificant" or "almost zero" would work). This has been corrected.

9- Refs. [22-24] should be cited between Ref. [16] and Ref. [17]. This has been corrected.

10- Double check casing in the references ("Mineral" [8], "K" [15], "Heisenberg" [17], ...). This has been corrected.

REFEREE 2:

  1. Introduction: I did not understand why only minor magnetocrystalline anisotropy is expected for Fe2+ in the tetrahedral coordination. Orbital moment of Fe2+ is not quenched, so the anisotropy is likely. Moreover, the authors themselves resort to the single-ion anisotropy in the Conclusions when they discuss the magnetic structure between TN1 and TN2. This inconsistency may confuse readers and should be avoided.

We agree with the referee that this inconsistency may confuse readers and have therefore reformulated the paragraph in the introduction.

  1. Fig. 8b: I could not see the "opposite triangular chirality between two T1 triangles separated by z~0.5", or perhaps I do not quite understand what the authors mean by the "triangular chirality" (is it scalar chirality? or vector chirality?). In my view, the spins rotate counter-clockwise when going around the triangle in the clockwise direction, and this kind of chirality is the same for all the T1 triangles shown in this figure. A clarification along these lines would be helpful.

Note that the part before the quoted passage says: "..., because a spin configuration which yields a 120$^\circ$ alignment on all triangles - which does not explain the experimental data - requires an opposite...", i.e. we refer to a different magnetic structure which is not in agreement with the data. The depicted magnetic structure reveals - as the referee correctly points out - a unique triangular chirality for the T1 triangles.

---

## Editorial Decision

published